# Viral Shedding among Re-Positive Severe Acute Respiratory Syndrome Coronavirus-2 Positive Individuals in Republic of Korea

**DOI:** 10.3390/v13102089

**Published:** 2021-10-17

**Authors:** Jeong-Min Kim, Boyeong Ryu, Young June Choe, Hye-Jun Jo, Hyeokjin Lee, Heui Man Kim, Nam-Joo Lee, Jee Eun Rhee, Yoon-Seok Chung, Myung-Guk Han, Eun-Jin Kim, Youngjoon Park, Jin Gwack, Yeowon Jin, Jeongsuk Song, Seunghee Seo, Byoungchul Gill, Hyunyeong Kim, Yeeun Park, Cheon Kwon Yoo, Eun Kyeong Jeong

**Affiliations:** 1Division of Emerging Infectious Diseases, Bureau of Infectious Disease Diagnosis Control, Korea Disease Control and Prevention Agency, Cheongju-si 28159, Korea; jmkim97@korea.kr (J.-M.K.); jhj0505@korea.kr (H.-J.J.); jinny0909@korea.kr (H.L.); animal80@korea.kr (H.M.K.); njlee@korea.kr (N.-J.L.); jerhee001@korea.kr (J.E.R.); ekim@korea.kr (E.-J.K.); 2Division of Risk Assessment, Korea Disease Control and Prevention Agency, Cheongju-si 28159, Korea; byryu@korea.kr; 3Department of Social and Preventive Medicine, Hallym University College of Medicine, Chuncheon-si 24252, Korea; choey@korea.ac.kr; 4Honam Regional Center for Disease Control and Prevention, Division of Infectious Disease Diagnosis Control, Korea Disease Control and Prevention Agency, Seo-gu, Gwangju-si 61947, Korea; rollstone93@korea.kr; 5Division of Viral Diseases, Bureau of Infectious Disease Diagnosis Control, Korea Disease Control and Prevention Agency, Cheongju-si 28159, Korea; mahan@korea.k; 6Director for Epidemiological Investigation Analysis, Korea Disease Control and Prevention Agency, Cheongju-si 28159, Korea; pahmun@korea.kr; 7Division of Emerging Infectious Disease Response, Bureau of Infectious Disease Emergency Preparedness and Response, Korea Disease Control and Prevention Agency, Cheongju-si 28159, Korea; gwackjin@korea.kr (J.G.); ywjin@korea.kr (Y.J.); 8Division of Infectious Disease Control, Bureau of Infectious Disease Policy, Korea Disease Control and Prevention Agency, Cheongju-si 28159, Korea; jssong1@korea.kr; 9Division of Laboratory Diagnosis Management, Bureau of Infectious Disease Diagnosis Control, Korea Disease Control and Prevention Agency, Cheongju-si 28159, Korea; lilyflo32@korea.kr (S.S.); hykim101@korea.kr (H.K.); biosafety08@korea.kr (Y.P.); 10Division of Bacterial Diseases, Bureau of Infectious Disease Diagnosis Control, Korea Disease Control and Prevention Agency, Cheongju-si 28159, Korea; gilri@korea.kr; 11Bureau of Infectious Disease Diagnosis Control, Korea Disease Control and Prevention Agency, Cheongju-si 28159, Korea; ckyoo@korea.kr; 12Korea Disease Control and Prevention Agency, Cheongju-si 28159, Korea

**Keywords:** viral shedding, re-positive, coronavirus disease, severe acute respiratory syndrome coronavirus 2

## Abstract

This study investigated the infectivity of severe acute respiratory syndrome (SARS-CoV-2) in individuals who re-tested positive for SARS-CoV-2 RNA after recovering from their primary illness. We investigated 295 individuals with re-positive SARS-CoV-2 polymerase chain reaction (PCR) test results and 836 of their close contacts. We attempted virus isolation in individuals with re-positive SARS-CoV-2 PCR test results using cell culture and confirmed the presence of neutralizing antibodies using serological tests. Viral culture was negative in all 108 individuals with re-positive SARS-CoV-2 PCR test results in whom viral culture was performed. Three new cases of SARS-CoV-2 infection were identified among household contacts using PCR. Two of the three new cases had had contact with the index patient during their primary illness, and all three had antibody evidence of past infection. Thus, there was no laboratory evidence of viral shedding and no epidemiological evidence of transmission among individuals with re-positive SARS-CoV-2 PCR test results.

## 1. Introduction

The coronavirus disease (COVID-19) pandemic is causing high morbidity and mortality worldwide. As of 24 June 2021, 178,837,204 confirmed cases and 3,880,450 deaths had been reported in 224 countries, areas, and territories [1]. During the same period, 153,155 cases were confirmed with 2008 deaths and 151,147 cases were discharged from isolation in Republic of Korea [2].

Although most COVID-19 patients recover with few complications after hospital discharge, some are recurrently positive on re-testing, leading to their re-isolation. If the SARS-CoV-2 in re-positive COVID-19 patients is not infectious, placing them in quarantine could divert essential healthcare resources away from other patients. In this context, we investigated the infectivity of SARS-CoV-2 in COVID-19 patients who were re-tested and found to be positive for sudden acute respiratory syndrome coronavirus 2 (SARS-CoV-2) after they had recovered from their primary illness.

## 2. Materials and Methods

### 2.1. Ethics Approval and Informed Consent

The study was approved by the KDCA Institutional Review Board (2020-03-01-P-A). The board waived the requirement for written informed consent.

### 2.2. Setting and Data Source

We analyzed data from patients who were reported to be recurrently positive between 10 April and 3 May 2020. From 10 April 2020, the Korea Disease Control and Prevention Agency (KDCA) started mandatory daily reporting of re-positive cases by e-mail. A re-positive case was defined as an individual with confirmed COVID-19 who was re-tested for SARS-CoV-2 after isolation was lifted, using a polymerase chain reaction (PCR), and tested positive. A confirmed COVID-19 patient was defined as an individual who had been confirmed to be infected with SARS-CoV-2 using PCR, and/or virus isolation, regardless of the clinical manifestations. The methods used for laboratory confirmation of SARS-CoV-2 complied with the World Health Organization guidelines [1]. All patients with confirmed COVID-19 were kept in mandatory isolation and were required to test negative twice on PCR from samples collected at least 24 h apart. To identify the infectivity of SARS-CoV-2 in re-positive individuals, we monitored the occurrence of secondary infection among the contacts of re-positive cases and conducted laboratory testing to determine the presence of SARS-CoV-2.

### 2.3. Re-Positive Case Management and Contact Tracing

Considering the uncertainty regarding the infectivity, the management of re-positive cases was the same as that of newly confirmed cases regarding isolation and contact tracing. Epidemiological investigation of re-positive cases was conducted in a community health center by interview. General characteristics, presence of symptoms in re-positive individuals, and the reason for testing were investigated. In addition, family members and other contacts who had contact with re-positive individuals within two days before the diagnosis among asymptomatic cases, or within two days before the recurrence of symptoms among the symptomatic cases, were investigated, and the occurrence of symptoms and the results of the confirmatory tests were monitored for at least 14 days. Most community health centers conducted laboratory tests to screen family contacts of re-positive individuals at the time of the re-positive test result and all community health centers conducted laboratory screening of family contacts before the end of the quarantine period. If a contact was confirmed positive on SARS-CoV-2 PCR testing, their history of exposure to risk factors and past symptoms were investigated by interview and hospital visit history, and virus culture and plaque reduction neutralization tests (PRNTs) were performed on the positive specimens from the contacts and the re-positive individuals.

### 2.4. Specimen Collection

Respiratory specimens, such as nasopharyngeal and oropharyngeal swabs, were collected from 108 re-positive individuals at the time of the re-positive test from 27 March 2020 to 3 May 2020. All 108 specimens were confirmed positive by quantitative real-time reverse transcription PCR (rRT-PCR) performed at the Institute of Health and Environment and private testing institutions and sent to KDCA to investigate viral infectivity by cell culture. All specimens were tested in a Class 2 or higher biosafety cabinet according to the KDCA COVID-19 guidelines.

### 2.5. RNA Extraction and Quantitative Real-Time Reverse Transcription Polymerase Chain Reaction

RNA extraction from nasopharyngeal and oropharyngeal swabs and sputum specimens and quantitative rRT-PCR were performed as described by Kim et al. [3]. RNA was extracted from 140 µL of the sample using a Qiagen Viral RNA Mini kit (Qiagen, Hilden, Germany) according to the manufacturer’s instructions. The cycle threshold (Ct) value of the extracted RNA was determined by performing rRT-PCR using primers and probes specific for target genes of COVID-19 RNA-dependent RNA polymerase and envelope.

### 2.6. Virus Isolation

The isolation of SARS-CoV-2 was performed using specimens of re-positive individuals who had been confirmed positive by rRT-PCR. For pretreatment, the specimens were mixed 4:1 with a 1:1 mixture of nystatin (10,000 units/mL) and penicillin-streptomycin (10,000 U/mL) and reacted for 1 h at 4 °C. Subsequently, the mixture was centrifuged, and the supernatant was used for inoculation. The monkey kidney-derived cell line Vero E6 and the human intestinal Caco-2 cell line were used and cultured in Dulbecco’s modified Eagle’s medium supplemented with 10 or 20% fetal bovine serum, respectively, and 1% penicillin at 37 °C in a 5% CO_2_ atmosphere. Each cell was seeded in a 12-well plate with 2 × 105 cells/well one day before inoculation. On the day of inoculation, 100 μL pretreated primary culture specimen were inoculated in a well with 900 µL 2% fetal bovine serum–Dulbecco’s modified Eagle’s medium, harvested after culturing for five days, and centrifuged at 3000 rpm for 10 min. The supernatant was retrieved after centrifugation. Secondary culture was performed in the same manner by inoculating wells with the supernatant of the primary culture, harvesting it after five days, and centrifuging the culture specimen at 3000 rpm for 10 min to retrieve the supernatant. Virus proliferation was confirmed by the observation of cytopathic effects and the result of SARS-CoV-2 rRT-PCR using RNA extracted from cell culture media. Virus isolation was performed in a Biosafety Level 3 laboratory.

## 3. Results

We retrospectively reviewed 295 cases reported as re-positive for SARS-CoV-2 and 836 of the close contacts of the case patients. The characteristics of the cases are shown in Table 1. Their average age was 45.2 years and 64.1% were female. At the time of the re-positive test, 55.6% were asymptomatic. The most common reasons for re-testing were either as a screening measure or because of persistent symptoms. Of the 234 individuals who were symptomatic at the time of their original diagnosis, the median time from symptom onset to the re-positive test was 45.2 days (range: 8–82 days) (Figure 1).

Virus isolation was attempted in 108 of the individuals with re-positive SARS-CoV-2 on PCR testing. All were culture-negative, and 79 (89%) had a quantitative rRT PCR Ct >30. Of the 105 individuals tested for respiratory viruses, four tested positive for adenovirus, and one tested positive for bocavirus (Table 2).

Among 836 close contacts of the individuals with re-positive SARS-CoV-2 positivity, three secondary cases were identified, resulting in a secondary attack rate of 0.4% (Table 3). All three cases were household contacts, and two had had contact with the index patient during their initial illness.

Figure 2 shows a case summary of the three sets of re-positive cases with secondary positive contacts. Positive Contact 1 (of Set 1) was tested because she had been exposed to Re-positive Case 1; however, she had prior symptoms and had a history of exposure to clusters of COVID-19 at a religious gathering in February and to another COVID-19 patient. The PRNTs of this pair decreased from 1:123 to 1:17 and from 1:34 to <1:10 over a two-week period in the index patient and the positive contact, respectively.

Positive Contact 2 had a history of exposure to two of the re-positive cases. The PRNTs of this set were high at 1:297, 1:123, and 1:158 for the two re-positive cases and the contact, respectively. Positive Contact 3 had exposure to Re-positive Case 3, but also had exposure to other individuals with confirmed COVID-19. Positive Contact 3 had equivocal PCR results on two separate sets of nasopharyngeal and oropharyngeal swab specimens. The PRNTs of this pair were low, at 1:40 and 1:26 for Re-positive Case 3 and Positive Contact 3, respectively. Viral culture was attempted for the first two sets of re-positive cases and positive contacts, but all cultures were negative.

## 4. Discussion

Generally, acute viral respiratory infections do not relapse in immunocompetent patients. However, the COVID-19 pandemic has created heightened awareness and raised concerns about the possibility of a relapse in individuals who have recovered from COVID-19. In 2003, during the severe acute respiratory syndrome outbreak in Hong Kong, there was a report of a 60-year-old woman who was discharged after a three-week hospital admission for pneumonia of unknown etiology, and was readmitted with respiratory symptoms and increased coronavirus antibody titer [4]. With SARS-CoV-2, a 46-year-old woman in China was reported as a “recurrent case” after having a positive test result on day 8 and day 17 [5]. Among 172 COVID-19 patients who were discharged after meeting the discharge criteria in China (improved clinical and laboratory values with at least two consecutive negative RT-PCR results on samples collected at least at 24 h apart), 25 (14.5%) were readmitted to hospital because of a reverted positive rRT-PCR result [6]. Most of these patients were asymptomatic, and only eight (32%) had mild cough.

SARS-CoV-2 may be shed from mucosal surfaces, particularly in the respiratory and gastrointestinal systems. A study of 191 COVID-19 patients (median age: 56 years) in China showed that the duration of viral shedding was 20 days (interquartile range: 17–24 days), with the longest observed duration of viral shedding in survivors to be 37 days [7]. A study in Taiwan detected virus on day 63 with virus isolation in the first 18 days after symptom onset [8]. SARS-CoV-2 was detectable until death in non-survivors, suggesting possible correlation between prolonged shedding and disease severity. The persistence of SARS-CoV-2 shedding suggests that there is a risk of transmitting infection to others after the end of quarantine. However, epidemiological studies have not shown evidence of infection among the contacts of individuals with persistent SARS-CoV-2 PCR positivity who were only exposed to a patient during the re-positive period. The recent advent of molecular diagnostic techniques commonly demonstrates prolonged shedding of the virus, even with clinical improvement. According to laboratory test results, the virus can be continuously detected through PCR tests even after antibodies are detected and virus isolation is impossible [8,9]. Indeed, PCR, the most widely used molecular method, does not differentiate between viable and non-viable viruses, thereby overestimating the persistence of viral shedding. Notwithstanding, a positive PCR test may or may not indicate the presence of viable virus, and only a positive virus culture confirms the potential for transmission. Based on the results of this study, we changed the management of re-positive cases and no longer conduct isolation and contact tracing of patients with re-positive SARS-CoV-2 reverse transcription PCR results or recommend quarantine for two weeks after discharge. By changing these measures, the responders reduced the burden of managing re-positive cases and could concentrate on preventing further spread in the community through newly confirmed patients of COVID-19. However, as the COVID-19 pandemic continues with increasing risk of re-infection due to emerging new variants, epidemiological factors and laboratory test results of re-positive cases are monitored to evaluate if there are cases of re-infection, not recurrence.

Rigorous mass testing likely has contributed to the success in flattening the curve of the COVID-19 epidemic in Korea, as of mid-April 2020. The goal of mass testing is to detect SARS-CoV-2 with high sensitivity to avoid missing any cases; however, diagnostic advances alone cannot achieve this goal. Despite its obvious public health importance, there has been little discussion on the importance of diagnostic stewardship during the COVID-19 pandemic. Informed decisions must be made about the conditions under which PCR re-testing is required, how the test results will be used, and whether re-testing is indicated. Epidemiological factors and symptoms could be considered with proper public health management.

In conclusion, we found that most patients with re-positive SARS-COV-2 after discharge were mild or asymptomatic and did not find epidemiological evidence of transmission by individuals after a re-positive test. We recommend diagnostic stewardship before performing re-testing of COVID-19 patients who have experienced a clinical recovery. Additionally, rapid assessment whether re-positive and re-infection for cases who tested positive after recovery of primary infection will be needed to determine the initiatives of respond properly.

## Figures and Tables

**Figure 1 viruses-13-02089-f001:**
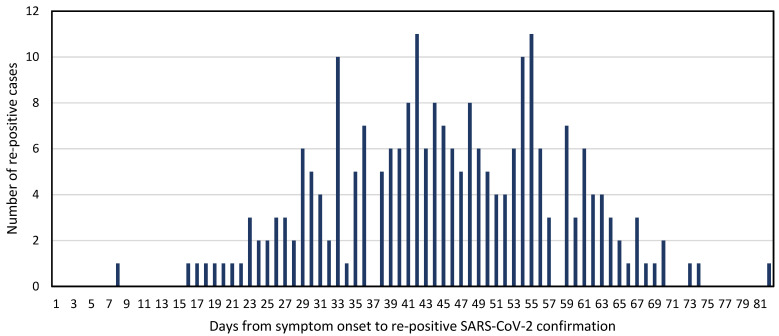
Time to confirmation of re-positive SARS-CoV-2 from initial symptom onset in Republic of Korea, 2020 (*n* = 234).

**Figure 2 viruses-13-02089-f002:**
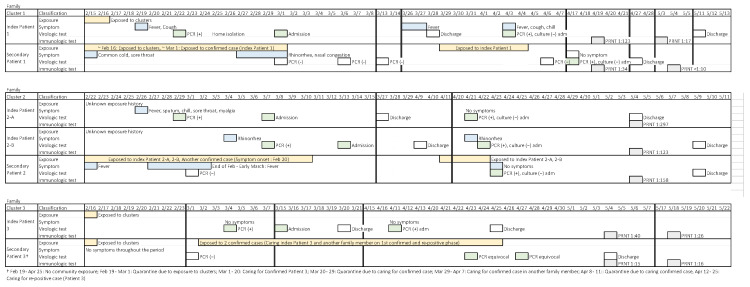
Case summary of secondary cases of SARS-CoV-2 positivity among contacts of patients with re-positive SARS-CoV-2. Abbreviations: Adm, admission; PCR, polymerase chain reaction; PRNT, plaque reduction neutralization test.

**Table 1 viruses-13-02089-t001:** Clinical characteristics of individuals with confirmed re SARS-CoV-2 positivity on PCR testing after recovery from COVID-19, Republic of Korea, 2020 (*n* = 295).

Characteristic	No.	(%)
Age group		
0–19 y	22	(7.5)
20–29 y	72	(24.4)
30–39 y	38	(12.9)
40–49 y	36	(12.2)
50–59 y	52	(17.6)
60–69 y	31	(10.5)
≥70 y	44	(14.9)
Sex		
Female	189	(64.1)
Male	106	(35.9)
Symptoms at time of re-testing (*n* = 294) ^1^		
Fever	29	(9.2)
Cough	55	(18.7)
Sputum	36	(12.2)
Rhinorrhea	19	(6.5)
Sore throat	33	(11.2)
Myalgia	25	(8.5)
Asymptomatic	163	(55.4)
Reason for re-testing		
Screening test	136	(46.1)
Persistent symptoms	111	(37.6)
New contact with a confirmed case	28	(9.5)
Hospital visit for another reason	12	(4.1)
Self-request	8	(2.7)

^1^ One individual without information on symptoms at the time of re-testing positive was excluded. Abbreviations: COVID-19, coronavirus disease; SARS-CoV-2, severe acute respiratory syndrome coronavirus 2.

**Table 2 viruses-13-02089-t002:** Laboratory results of individuals with confirmed re-positive SARS-CoV-2 on PCR testing after recovery from COVID-19, Republic of Korea, 2020 (*n* = 108).

Test	No.	(%)
Virus culture		
Negative	108	(100.0)
Ct value (*n* = 79) ^1^		
25–30	8	(10.1)
≥30	71	(89.9)
Respiratory viruses (*n* = 105) ^2^		
Negative	100	(95.2)
Adenovirus	4	(3.8)
Bocavirus	1	(1.0)

^1^ Ct value at the time of the re-positive test; ^2^ samples were tested for influenza virus (A, B), human respiratory syncytial virus, human metapneumovirus, human parainfluenza virus, human adenovirus, bocavirus, rhinovirus, and human coronavirus. Abbreviations: Ct, cycle threshold; COVID-19, coronavirus disease; SARS-CoV-2, severe acute respiratory syndrome coronavirus 2.

**Table 3 viruses-13-02089-t003:** Results of contact monitoring of individuals with confirmed re-positive SARS-CoV-2 on PCR testing after recovery from COVID-19, Republic of Korea, 2020.

	Contacts	Secondary Cases	
	No. (%)	No.	Secondary Attack Rate (%)
Type of contact			
Household	363 (43.4)	3	(0.8)
Non-household	473 (56.6)	0	(0.0)
Symptomatic			
Yes	454 (54.3)	2	(0.4)
No	382 (45.7)	1	(0.3)
Total	836 (100.0)	3	(0.4)

A secondary case was defined as a contact who tested positive for SARS-CoV-2 on PCR. Abbreviations: COVID-19, coronavirus disease; SARS-CoV-2, severe acute respiratory syndrome coronavirus 2.

## Data Availability

The datasets used and/or analyzed during the current study are available from the corresponding author on reasonable request.

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
