# Peer review of "Viral Shedding among Re-Positive Severe Acute Respiratory Syndrome Coronavirus-2 Positive Individuals in Republic of Korea"

_viruses, 2021, doi:10.3390/v13102089_

Round 1

Reviewer 1 Report

Thank you for this work. It will add usefully to the literature. A few comments follow.

  • People are not infectious, pathogens are. Please be mindful of syntax regarding infectivity. For instance, the first sentence of the abstract could be adjusted to, "This study investigated the viral burden in secretions of individuals with coronavirus disease who re-tested positive for severe acute respiratory syndrome coronavirus 2 (SARS-CoV-2) RNA after recovering from their primary illness." Please make this or a similar adjustment through-out. For instance, it happens again on lines 58 and 78.
  • I am struggling to understand a syntax versus semantic issue, in part because you used the phrase primary infection rather than primary illness. When you say recurrence, are you asserting that they (1) have become infected again from a new exposure, (2) have recrudescent disease, or (3) either? Those within 30-45 days or with concomitant illnesses are quite reasonable for (2) and those greater than 45 days are suggestive of (1) > (2). This has implications for your premise/ structure and analysis and should be clarified, analyses adjusted.
  • It is possible that one of the reasons observed attack rate of household contacts was low was because they had COVID-19 commensurate with the enrolled cases' primary illnesses. It appears that you only assessed neutralizing Ab on those contacts who were +?
  • What were the cycle times of the cases purportedly sources of household contact infections? 

Reviewer 2 Report

1. Is there possibility that recurrent positive cases were missed? or all recovered cases were re-tested? 2. Possibility of transmission of infection via SARS-CoV2 viruses in stool if the individuals with re-infection detected stool PCR positivity? 3. Apart from diagnostic stewardship, any recommendations from this study generated? e.g. re-infection patients isolation requirement? 4. Any difference in the conclusion in era of SARS-CoV2 variants?
